# Dose-dependent oral glucocorticoid cardiovascular risks in people with immune-mediated inflammatory diseases: A population-based cohort study

**Mar Pujades-Rodriguez**[1☯]*, **Ann W. Morgan**[2,3], **Richard M. Cubbon**[2], **Jianhua Wu**[4☯]

**1** Leeds Institute of Health Sciences, School of Medicine, University of Leeds, Leeds, United Kingdom, **2** Leeds Institute of Cardiovascular and Metabolic Medicine, School of Medicine, University of Leeds, Leeds, United Kingdom, **3** NIHR Biomedical Research Centre, Leeds Teaching Hospitals NHS Trust, Chapel Allerton Hospital, Leeds, Leeds, United Kingdom, **4** School of Dentistry, University of Leeds, Leeds, United Kingdom

☯ These authors contributed equally to this work.
\* M.D.M.PujadesRodriguez@leeds.ac.uk

**Data Availability Statement:** Data cannot be shared publicly because of signed Sharing Data Confidential Agreements. Access to raw data can

## Abstract

### Background

Glucocorticoids are widely used to reduce disease activity and inflammation in patients with a range of immune-mediated inflammatory diseases. It is uncertain whether or not low to moderate glucocorticoid dose increases cardiovascular risk. We aimed to quantify glucocorticoid dose-dependent cardiovascular risk in people with 6 immune-mediated inflammatory diseases.

### Methods and findings

We conducted a population-based cohort analysis of medical records from 389 primary care practices contributing data to the United Kingdom Clinical Practice Research Datalink (CPRD), linked to hospital admissions and deaths in 1998–2017. We estimated time-variant daily and cumulative glucocorticoid prednisolone-equivalent dose-related risks and hazard ratios (HRs) of first all-cause and type-specific cardiovascular diseases (CVDs). There were 87,794 patients with giant cell arteritis and/or polymyalgia rheumatica (*n* = 25,581), inflammatory bowel disease (*n* = 27,739), rheumatoid arthritis (*n* = 25,324), systemic lupus erythematosus (*n* = 3,951), and/or vasculitis (*n* = 5,199), and no prior CVD. Mean age was 56 years and 34.1% were men. The median follow-up time was 5.0 years, and the proportions of person–years spent at each level of glucocorticoid daily exposure were 80% for non-use, 6.0% for <5 mg, 11.2% for 5.0–14.9 mg, 1.6% for 15.0–24.9 mg, and 1.2% for ≥25.0 mg.

Incident CVD occurred in 13,426 (15.3%) people, including 6,013 atrial fibrillation, 7,727 heart failure, and 2,809 acute myocardial infarction events. One-year cumulative risks of all-cause CVD increased from 1.4% in periods of non-use to 8.9% for a daily prednisolone-equivalent dose of ≥25.0 mg. Five-year cumulative risks increased from 7.1% to 28.0%, respectively. Compared to periods of non-glucocorticoid use, those with <5.0 mg daily

be requested from the CPRD (https://cprd.com) providing the study approval reference (16_146).

**Funding:** A.W.M. is supported by the Medical Research Council TARGET Partnership Grant (Treatment According to Response in Giant cEll arTeritis) (MR/N011775/1) and the National Institute for Health Research (NIHR) Medtech and In vitro Diagnostics Co-operatives at Leeds (MIC-2016-015). A.W.M. and J.W. are supported by the NIHR Biomedical Research Centre at Leeds (IS-BRC-1215-20015). The views expressed are those of the author(s) and not necessarily those of the National Health Service, the NIHR or the Department of Health and Social Care. The funders had no role in study design, data collection and analysis, decision to publish, or preparation of the manuscript.

**Competing interests:** I have read the journal's policy and the authors of this manuscript have the following competing interests: Dr. AWM reports grants from Medical Research Council, grants from National Institute for Health Research, during the conduct of the study; grants and personal fees from Roche/ Chugai, personal fees from Sanofi, personal fees from GlaxoSmithKline, personal fees from Regeneron, outside the submitted work. Dr. MPR is currently employed by IQVIA (start of contract >1 year after completing and submitting this work). It is not possible to spell out IQVIA because there is no long form for it: https://www.quora.com/What-is-the-full-form-of-IQVIA

**Abbreviations:** BMI, body mass index; CI, confidence interval; COPD, chronic obstructive pulmonary disease; CPRD, Clinical Practice Research Datalink; CVD, cardiovascular disease; HR, hazard ratio; IQR, interquartile range; ISAC, Independent Scientific Advisory Committee for Medicines and Healthcare products Regulatory Agency database research; RECORD-PE, reporting of studies conducted using observational routinely collected health data statement for pharmacoepidemiology; SD, standard deviation.

prednisolone-equivalent dose had increased all-cause CVD risk (HR = 1.74; 95% confidence interval [CI] 1.64–1.84; range 1.52 for polymyalgia rheumatica and/or giant cell arteritis to 2.82 for systemic lupus erythematosus). Increased dose-dependent risk ratios were found regardless of disease activity level and for all type-specific CVDs. HRs for type-specific CVDs and <5.0-mg daily dose use were: 1.69 (95% CI 1.54–1.85) for atrial fibrillation, 1.75 (95% CI 1.56–1.97) for heart failure, 1.76 (95% CI 1.51–2.05) for acute myocardial infarction, 1.78 (95% CI 1.53–2.07) for peripheral arterial disease, 1.32 (95% CI 1.15–1.50) for cerebrovascular disease, and 1.93 (95% CI 1.47–2.53) for abdominal aortic aneurysm.

The lack of hospital medication records and drug adherence data might have led to underestimation of the dose prescribed when specialists provided care and overestimation of the dose taken during periods of low disease activity. The resulting dose misclassification in some patients is likely to have reduced the size of dose–response estimates.

## Conclusions

In this study, we observed an increased risk of CVDs associated with glucocorticoid dose intake even at lower doses (<5 mg) in 6 immune-mediated diseases. These results highlight the importance of prompt and regular monitoring of cardiovascular risk and use of primary prevention treatment at all glucocorticoid doses.

## Author summary

### Why was this study done?

- Glucocorticoids (steroids) are widely used to reduce disease activity and inflammation in patients with a range of immune-mediated inflammatory diseases, such as rheumatoid arthritis, polymyalgia rheumatica, giant cell arteritis, and inflammatory bowel disease.

- Adequate assessment of cost-effectiveness of new steroid-sparing treatments for immune and inflammatory diseases require modelling of estimates of risk and cost of the main treatment complications of steroids.

- It is widely recognised that high-dose steroids may increase the risk of cardiovascular disease (CVD; heart disease, stroke, or other vascular diseases), but it is debated whether this increase also applies to lower steroid doses.

- Earlier studies of CVD risk associated with glucocorticoid therapy failed to account for changes in dose over time and for use of non-oral steroids and other potentially confounding therapies.

### What did the researchers do and find?

- In 87,794 adults with immune-mediated inflammatory diseases and no prior CVD (5-year median follow-up), we studied the risk of 6 common CVDs associated with the steroid dose prescribed, quantified either as current or as cumulative dose.

- We found strong dose-dependent risks of all CVDs, including myocardial infarction, heart failure, atrial fibrillation, and cerebrovascular disease, in patients diagnosed with the 6 inflammatory diseases studied.

- After 1 year, the overall absolute risk of CVD doubled for individuals using less than 5 mg prednisolone per day and was 6 times higher for users of 25 mg or greater.

- Many individuals had known modifiable cardiovascular risk factors, including current smoking (24%), obesity (25%), or hypertension (25%).

### What do these findings mean?

- We have provided evidence that individuals receiving steroids have an increased risk of developing a broad spectrum of fatal and nonfatal CVDs and that this risk increases with the dose of steroids and with the duration of steroid treatment.

- It was previously believed that less than 5 mg of prednisolone was safe long term, but even at this "low dose" patients with immune-mediated inflammatory diseases have a doubling of their underlying risk of CVD.

- New treatment approaches that avoid the need for long-term steroid treatment and have better cardiovascular safety profile are required for immune-mediated inflammatory diseases.

- All patients requiring long-term steroid treatment should be prescribed the lowest effective steroid dose and have a personalised CVD risk prevention plan that takes into account current and prior steroid use.

## Introduction

Patients with immune-mediated inflammatory diseases often receive courses of oral glucocorticoids to reduce disease activity and inflammation during the initial episode and subsequent episodic flares. Prolonged glucocorticoid treatment often causes adverse events, including cardiovascular diseases (CVDs) [1–3]. Glucocorticoids can increase cardiovascular risk through direct and indirect metabolic syndrome enhancement [4–6] and mineralocorticoid effects, including cellular membrane electrolyte-mediated efflux [7–9]. However, the anti-inflammatory and immune-suppressive effects of glucocorticoids can also lower or neutralise the atherosclerotic and vascular injury effects of chronic inflammatory diseases [10]. Demonstration of cost-effectiveness of newly licenced glucocorticoid-sparing drugs, such as biologics, is critical to guide their introduction for the treatment of immune-mediated inflammatory diseases in routine healthcare. These studies require the accurate estimation of glucocorticoid dose–response relationships to quantify cost savings associated with the toxicity profile of new drugs.

Evidence of the relationship between glucocorticoids and CVDs comes primarily from studies of associations with current baseline medication use or dose [2,3,7,11–16], ignoring the doses previously administered and their changes over time, as well as the concomitant use of other common medications that can affect the risk of CVDs (e.g., nonsteroidal anti-inflammatory drugs). Many also have failed to adjust for important cardiovascular risk factors, such as

smoking [1,2,7,13,16]. These studies have reported a dose-dependent risk of CVD with weaker associations for daily prednisolone-equivalent doses lower than 5 to 10 mg[2,3,13,17].

Our study aimed to estimate daily and cumulative dose-dependent oral glucocorticoid cardiovascular disease risk in people diagnosed with 6 common immune-mediated inflammatory diseases in England, using time-dependent regression methods.

## Methods

### Setting and data sources

We analysed linked electronic health records from people registered at family practices in the Clinical Practice Research Datalink (CPRD) between 1 January 1998 and 15 March 2017. CPRD contains demographic and lifestyle data, diagnoses (e.g., stroke), prescribed medication and results of laboratory tests and clinical examinations, prospectively recorded during primary care contacts [18]. Previous validation studies have provided evidence of the accuracy of diagnostic and prescribing data [18]. Patients are broadly representative of the United Kingdom population with regard to age, sex, and ethnicity [18]. CPRD data were linked to hospital records and the mortality registry (S1 Text). Hospital records from the Hospital Episode Statistics (HES) (www.hscic.gov.uk/hes) contain diagnoses recorded during elective and emergency hospital admission across all National Health Service hospitals in England. Mortality data from the Office of National Statistics (ONS) (https://www.ons.gov.uk/atoz?query= mortality&size=10) were used to identify dates and causes of death.

### Ethical considerations

The study was approved by the Independent Scientific Advisory Committee for Medicines and Healthcare products Regulatory Agency database research (ISAC), reference 16_146.

### Study design and follow-up

This was a cohort study including all patients continuously registered in a CPRD practice for 1 year or more, aged ≥18 years, and free of CVD, who had been diagnosed with at least 1 of 6 immune-mediated inflammatory diseases commonly treated with oral glucocorticoids at or before the start of follow-up (inclusion eligibility criteria). These were polymyalgia rheumatica, giant cell arteritis, systemic lupus erythematosus, rheumatoid arthritis, vasculitis, and inflammatory bowel disease (Fig A in S1 Fig). Diagnostic codes used to identify patients with each immune-mediated inflammatory disease are shown in Table A in S1 Table. For each patient, the follow-up started when they first became eligible (i.e., earliest date on which all the inclusion criteria were met). It ended on the earliest of the following dates: occurrence of the outcome analysed (e.g., stroke), leaving the family practice, death, or last data collection date. For the combined analyses, patients were assigned to the immune-mediated inflammatory disease group corresponding to the earliest condition diagnosed in the database.

### Oral glucocorticoid exposure

For each prescription of oral glucocorticoids issued to the patients between 1 year before the start and the end of the follow-up dates, recorded in CPRD, we derived the daily dose from the recorded product name, which included information on product strength (e.g., 2 mg), directions given (e.g., 1 tablet once a day), and quantity prescribed (e.g., 28 tablets). We then estimated the duration of each oral glucocorticoid prescription dividing the quantity of tablets prescribed by the daily dose. Given the variation in relative anti-inflammatory effects of

different types of glucocorticoids, for each prescription, we finally converted the daily dosage into milligrams of prednisolone-equivalent dose (Table B in S1 Table).

We defined several time-variant glucocorticoid variables to quantify current and cumulative drug exposure: (1) ever use from 1 year (2 or 5 years in additional sensitivity analyses) prior to follow-up start (binary variable); (2) current daily use (i.e., whether or not the patient was prescribed glucocorticoids at a given time point [binary variable]); (3) current daily dose per 5 mg/day with 0 value when medication was not prescribed (continuous and categorical variables: non-use, 1 to 4.9 mg, 5.0 to 14.9 mg, 15.0 to 24.9 mg, $\geq$25.0 mg/day); and (4) cumulative dose since 1 year (2 or 5 years in additional sensitivity analyses) prior to follow-up start per 1,000 mg (i.e., sum of the total dose prescribed up to that point divided by 1,000; considered as continuous and categorical variables: non-use, 1 to 959 mg, 960 to 3,054 mg, 3,055 to 7,299 mg, and $\geq$7,300 mg; as defined previously [19–22]).

## Outcome measures

The primary outcome was the first occurrence of a composite of fatal and nonfatal CVDs (all-cause CVD). Secondary outcomes were the first occurrence of the following common types of CVDs: atrial fibrillation, heart failure, myocardial infarction, cerebrovascular disease, peripheral arterial disease, and abdominal aortic aneurysm. Diagnostic codes [23] used to define the outcomes are listed in Table C in S1 Table and have been validated and used in multiple previous studies [24–28].

## Confounding variables

We considered the following variables as a priori confounders: baseline age, sex, ethnicity, socioeconomic status (index of multiple deprivation [26,29], area-based indicator linked through the patient's home postcode), smoking status, body mass index (BMI), biomarkers (total, high- and low- density lipoprotein-cholesterol, systolic blood pressure, c-protein reactive protein, and creatinine), underlying disease (e.g., rheumatoid arthritis), comorbidities recorded in primary or hospital care (diabetes, diagnosed hypertension, cancer, asthma, chronic obstructive pulmonary disease [COPD], and renal disease), prescribed non-oral glucocorticoid medication (inhaled, nasal, parenteral/intra-articular, topical, and rectal), and the number of hospital visits 1 year before baseline. Continuous biomarker variables were included as cubic spline in the models. We also considered time-variant prescribed medication (disease-modifying antirheumatic drugs and nonsteroidal anti-inflammatory drugs) during follow-up. Detailed definition of covariates is shown in S1 Text.

## Statistical analysis

We replaced missing daily dose of oral glucocorticoids (i.e., during tapering periods) and confounders through multiple imputation with chained equations with generation of 25 datasets (S1 Text). Models for dose imputation included patient demographics (i.e., age, sex, and index of multiple deprivation [30]), underlying immune-mediated inflammatory disease, time between follow-up start and prescription, type of oral glucocorticoid (e.g., prednisone), and diagnosed comorbidities (e.g., diabetes).

We used standard descriptive statistics to describe baseline patient characteristics. We estimated cumulative probabilities of CVD outcomes using cumulative incidence functions to prevent the introduction of bias associated with the presence of competing risk of deaths [31]. We calculated incident rates with 95% confidence intervals (95% CIs) dividing the number of patients with incident CVD by the total number of person–years of follow-up.

We assessed the association between the outcomes and each of the oral glucocorticoid exposures using Cox proportional hazards models adjusted for the a priori confounders, with the practice identifier included as a random intercept to account for clustering effect. No interaction terms were included. The proportional hazards assumption was assessed using Schoenfeld residuals tests. The primary analysis was based on covariate-imputed data. We generated models for each of the 25 imputed datasets and pooled estimates and accompanying 95% CIs following Rubin's rules. We used 2-sided tests and considered significant at $p < 0.05$. We performed the data management in Stata (StataCorp LP, College Station, United States of America; version 15) and analyses in R (http://cran.r-project.org/ The R Foundation for Statistical Computing, Austria; version 3.3.1).

In secondary analyses, we modelled cardiovascular risk separately for men and women, for each of the 6 immune-mediated inflammatory diseases studied, and according to duration of these diseases at the start of follow-up (newly diagnosed/incident, within 2 years and over 2 years since diagnosis).

In sensitivity analyses, we obtained estimates from complete case models (i.e., restricted to patients with complete covariate data), from models including covariates with a separate category for missing data and from models unadjusted for biomarker data with a level of missingness >60%. We also additionally adjusted the models for the level of disease activity. We defined periods of active disease based on c-reactive protein and erythrocyte sedimentation rate levels ($\geq$10 mg/mL and $\geq$30 mm/h, respectively) and the glucocorticoid daily dose (increase in prednisolone-equivalent dose by >5 or 10 mg that was sustained for over 3 weeks) (S1 Text).

MPR and JW had full access to all the data used in the study. All the analyses were planned except for the sensitivity analyses additionally performed in response to peer review comments to test the robustness of the primary analyses. These were: (1) consider longer periods of prescribing prior to the index date (i.e., 2 years and 5 years) to control for protopathic bias (as CVD is associated both with drug exposure and the indication for the exposure); and (2) propensity score adjustment of Cox proportional hazard models to balance the covariates between exposure groups, in order to control for residual confounding by indication (details of implementation are provided in S1 Text). A summary of the generic protocol to study safety (i.e., risks of different types of adverse events) and associated costs of glucocorticoid therapy for the treatment of chronic inflammatory diseases is available at https://www.cprd.com/protocol/safety-and-associated-costs-glucocorticoid-therapy-treatment-chronic-inflammatory-diseases. No specific protocol was written for the analyses of cardiovascular risk. This study is reported as per the reporting of studies conducted using observational routinely collected health data statement for pharmacoepidemiology (RECORD-PE) guideline (S1 Checklist).

## Results

### Patient characteristics

The study included 87,794 adults from 389 general practices with at least 1 immune-mediated inflammatory disease diagnosed; 25,581 with polymyalgia and/or giant cell arteritis, 27,739 (31.6%) with inflammatory bowel disease, 25,324 (28.8%) with rheumatoid arthritis, 5,199 (5.9%) with vasculitis, and 3,951 (4.5%) with systemic lupus erythematosus (Table 1). A total of 20,851 (23.7%) patients transferred out of their general practices and were lost to follow-up. The overall mean age was 56 years (standard deviation (SD) 18.3), 29,935 (34.1%) were men, and 21,264 (24.2%) were current smokers. At baseline, the mean duration since immune-mediated disease diagnosis was 9.6 years (SD = 8.7; range from 6.8 years for polymyalgia and/

**Table 1. Patient baseline characteristics by type of immune-mediated inflammatory disease.**

| | All immune-mediated diseases | Polymyalgia rheumatica and/or giant cell arteritis | Inflammatory bowel disease | Rheumatoid arthritis | Systemic lupus erythematosus | Vasculitis |
|---|---|---|---|---|---|---|
| | N = 87,794 | N = 25,581 | N = 27,739 | N = 25,324 | N = 3,951 | N = 5,199 |
| **Follow-up time (years), mean (SD)** | 5.9 (4.7) | 5.6 (4.3) | 6.4 (5.0) | 6.2 (4.8) | 6.2 (4.9) | 5.5 (4.4) |
| Total | 521,161 | 142,216 | 176,547 | 156,846 | 24,567 | 28,343 |
| Since first recorded disease code | 9.6 (8.7) | 6.8 (5.3) | 11.7 (9.9) | 10.6 (9.4) | 10.3 (8.4) | 7.7 (6.7) |
| **Sociodemographic information** | | | | | | |
| **Males**, n (%) | 29,935 (34.1) | 7,210 (28.2) | 12,992 (46.8) | 6,963 (27.5) | 599 (15.2) | 2,171 (41.8) |
| **Age** (years), median [IQR] | 58.00 [41.00, 71.00] | 72.00 [65.00, 79.00] | 41.00 [31.00, 55.00] | 57.00 [46.00, 68.00] | 46.00 [35.00, 56.00] | 50.00 [35.00, 64.00] |
| **Ethnicity**, n (%) | | | | | | |
| White | 75,569 (86.1) | 22,371 (87.5) | 24,493 (88.3) | 21,478 (84.8) | 2,942 (74.5) | 4,285 (82.4) |
| Asian | 2,736 (3.1) | 376 (1.5) | 1,025 (3.7) | 855 (3.4) | 278 (7.0) | 202 (3.9) |
| Black | 1,042 (1.2) | 142 (0.6) | 251 (0.9) | 345 (1.4) | 238 (6.0) | 66 (1.3) |
| Other | 1,108 (1.3) | 158 (0.6) | 443 (1.6) | 312 (1.2) | 129 (3.3) | 66 (1.3) |
| **Index of multiple deprivation**, n (%) | | | | | | |
| First (least deprived) | 15,943 (18.2) | 5,080 (19.9) | 5170 (18.6) | 4,042 (16.0) | 662 (16.8) | 989 (19.0) |
| Fifth (most deprived) | 14,356 (16.4) | 3,180 (12.4) | 4,583 (16.5) | 4,945 (19.5) | 809 (20.5) | 839 (16.1) |
| **Biomarkers**, mean (SD) or median [IQR] | | | | | | |
| BMI (kg/m$^2$) | 26.8 (5.8) | 27.5 (5.6) | 25.7 (5.6) | 27.5 (6.1) | 26.2 (5.6) | 27.8 (6.6) |
| C-reactive protein (mg/L) | 9.00 [4.00, 28.00] | 16.80 [5.40, 45.00] | 5.05 [2.70, 16.00] | 8.50 [4.00, 22.00] | 4.30 [2.00, 8.00] | 5.00 [3.00, 16.00] |
| Erythrocyte sedimentation rate (mm/h) | 24.00 [10.00, 45.00] | 34.00 [17.00, 58.00] | 13.00 [6.00, 28.00] | 21.00 [10.00, 38.00] | 14.00 [7.00, 29.00] | 13.00 [5.00, 27.00] |
| Total cholesterol (mmol/L) | 5.2 (1.2) | 5.2 (1.2) | 5.0 (1.1) | 5.2 (1.1) | 5.1 (1.2) | 5.1 (1.2) |
| HDL cholesterol (mmol/L) | 1.5 (0.5) | 1.5 (0.5) | 1.4 (0.4) | 1.4 (0.5) | 1.5 (0.5) | 1.4 (0.5) |
| LDL cholesterol (mmol/L) | 3.06 (1.0) | 3.07 (1.0) | 3.0 (1.0) | 3.10 (1.1) | 3.05 (1.0) | 3.06 (1.1) |
| Haemoglobin (mmol/L) | 13.1 (1.5) | 12.9 (1.4) | 13.3 (1.7) | 13.0 (1.5) | 13.0 (1.5) | 13.5 (1.5) |
| Creatinine (mmol/L) | 78.00 [67.00, 90.00] | 80.00 [69.00, 94.00] | 78.00 [67.00, 90.00] | 75.00 [65.00, 87.00] | 74.00 [64.00, 85.00] | 79.00 [67.00, 94.00] |
| Systolic blood pressure (mmHg) | 134 (19.3) | 141 (18.1) | 126 (17.5) | 134 (18.9) | 127 (18.3) | 132 (18.5) |
| Diastolic blood pressure (mmHg) | 78 (10.1) | 79 (9.9) | 76 (10.1) | 79 (10.2) | 77 (10.3) | 78 (10.4) |
| **Health behaviour** | | | | | | |
| **Smoking status**, n (%) | | | | | | |
| Non-smoker | 32,995 (37.6) | 11,559 (45.2) | 9,361 (33.7) | 8,849 (34.9) | 1,201 (30.4) | 2,025 (38.9) |
| Ex-smoker | 11,654 (13.3) | 4,791 (18.7) | 3,079 (11.1) | 2,963 (11.7) | 243 (6.2) | 578 (11.1) |
| Current smoker | 21,264 (24.2) | 5,183 (20.3) | 6,630 (23.9) | 7,222 (28.5) | 1,000 (25.3) | 1,229 (23.6) |
| **No. of hospitalisation in last year**, median [IQR] | 0.00 [0.00, 1.00] | 0.00 [0.00, 0.00] | 0.00 [0.00, 1.00] | 0.00 [0.00, 1.00] | 0.00 [0.00, 1.00] | 0.00 [0.00, 1.00] |
| **Comorbidities**, n (%) | | | | | | |
| Diabetes | 5,602 (6.4) | 2,351 (9.2) | 1,107 (4.0) | 1,596 (6.3) | 181 (4.6) | 367 (7.1) |
| Hypertension | 22,072 (25.1) | 10,810 (42.3) | 3,251 (11.7) | 5,960 (23.5) | 736 (18.6) | 1,315 (25.3) |
| Asthma | 12,858 (14.6) | 3,597 (14.1) | 4,322 (15.6) | 3,567 (14.1) | 492 (12.5) | 880 (16.9) |
| Chronic obstructive pulmonary disease | 2,950 (3.4) | 1,280 (5.0) | 498 (1.8) | 999 (3.9) | 59 (1.5) | 114 (2.2) |

(*Continued*)

**Table 1.** (Continued)

| | All immune-mediated diseases | Polymyalgia rheumatica and/or giant cell arteritis | Inflammatory bowel disease | Rheumatoid arthritis | Systemic lupus erythematosus | Vasculitis |
|---|---|---|---|---|---|---|
| | N = 87,794 | N = 25,581 | N = 27,739 | N = 25,324 | N = 3,951 | N = 5,199 |
| Cancer | 5,005 (5.7) | 2,259 (8.8) | 998 (3.6) | 1,293 (5.1) | 154 (3.9) | 301 (5.8) |
| Renal disease | 1,095 (1.2) | 556 (2.2) | 171 (0.6) | 247 (1.0) | 36 (0.9) | 85 (1.6) |
| **Prescribed medication at baseline**, *n* (%) | | | | | | |
| Statins | 8,516 (9.7) | 4,437 (17.3) | 1,187 (4.3) | 2,198 (8.7) | 235 (5.9) | 459 (8.8) |
| Any blood pressure lowering medication | 23,755 (27.1) | 11,495 (44.9) | 3,645 (13.1) | 6,445 (25.5) | 802 (20.3) | 1,368 (26.3) |
| Angiotensin converting enzyme inhibitors | 8,151 (9.3) | 4,075 (15.9) | 1,215 (4.4) | 2,145 (8.5) | 226 (5.7) | 490 (9.4) |
| Angiotensin II receptor antagonist | 3,633 (4.1) | 1,960 (7.7) | 471 (1.7) | 853 (3.4) | 118 (3.0) | 231 (4.4) |
| **Prescribed medication in last year**, *n* (%) | | | | | | |
| Oral glucocorticoids | 14,356 (16.4) | 4,594 (18.0) | 4,583 (16.5) | 4,945 (19.5) | 809 (20.5) | 839 (16.1) |
| Inhaled or nasal glucocorticoids | 13,331 (15.2) | 4,523 (17.7) | 3,581 (12.9) | 3,895 (15.4) | 474 (12.0) | 858 (16.5) |
| Intramuscular or intra-articular glucocorticoids | 1,216 (1.4) | 470 (1.8) | 121 (0.4) | 584 (2.3) | 22 (0.6) | 19 (0.4) |
| Rectal glucocorticoids | 5,800 (6.6) | 531 (2.1) | 4,601 (16.6) | 491 (1.9) | 61 (1.5) | 116 (2.2) |
| Topical glucocorticoids | 2,094 (2.4) | 647 (2.5) | 615 (2.2) | 525 (2.1) | 139 (3.5) | 168 (3.2) |
| Nonsteroidal anti-inflammatory drugs | 39,690 (45.2) | 13,998 (54.7) | 5,143 (18.5) | 17,694 (69.9) | 1,371 (34.7) | 1,484 (28.5) |
| DMARDs ever during follow-up | 18,877 (21.5) | 1,112 (4.3) | 4,072 (14.7) | 12,147 (48.0) | 1,234 (31.2) | 316 (6.1) |

BMI, body mass index; DMARDs, disease modifying anti-rheumatic drugs; HDL-cholesterol, high-density lipoprotein-cholesterol; IQR, interquartile range; LDL-cholesterol, low-density lipoprotein-cholesterol; SD, standard deviation.

Continuous variables were presented as mean (SD) if they were normally distributed, and median [IQR] if they were not normally distributed. Data on ethnicity, BMI, c-reactive protein, erythrocyte sedimentation rate, total cholesterol, LDL-cholesterol, HDL-cholesterol, creatinine, systolic blood pressure, and smoking status were missing for 8.4%, 59.1%, 68.4%, 58.2%, 74.9%, 80.4%, 84.4%, 47.7%, 36.6%, and 24.9% of patients, respectively.

or giant cell arteritis and 11.7 years for inflammatory bowel disease). The most common patient comorbidities were hypertension (25.1%), asthma (14.6%), and diabetes (6.4%).

In the year prior to follow-up start, 14,356 (16.4%) patients were prescribed oral glucocorticoids, 13,331 (15.2%) inhaled or nasal glucocorticoids, and 39,690 (45.2%) nonsteroidal anti-inflammatory drugs. During follow-up, 18,877 (21.5%; range 4.2% for polymyalgia and/or giant cell arteritis to 48.0% for rheumatoid arthritis) received disease-modifying antirheumatic drugs.

## Incidence of fatal and nonfatal cardiovascular diseases

The median time of follow-up per patient was 5.0 (interquartile range (IQR) 2.0 to 6.2) years, and the proportions of person–years spent at each level of glucocorticoid daily exposure were 80% for non-use, 6.0% for <5 mg, 11.2% for 5.0 to 14.9 mg, 1.6% for 15.0 to 24.9 mg, and 1.2% for ≥25.0 mg. A total of 13,426 incident cardiovascular events occurred (15.3% of patients) over 541,655 person–years of follow-up (Table D in S1 Table), including 6,013 episodes of atrial fibrillation, 4,727 of heart failure, and 2,809 of acute myocardial infarction. The incidence of all-cause CVD was 24.8 per 1,000 person–years (95% CI 24.4 to 25.2). It increased from 18.5 (95% CI 18.1 to 18.9) per 1,000 person–years for periods of non-glucocorticoid use

to 45.6 (95% CI 42.1 to 49.2) for periods of ≥25 mg daily dose, and from 19.9 (95% CI 19.3 to 20.5) for unexposed periods to 26.4 (95% CI 25.5 to 27.2) per 1,000 person–years for ≥7,300 mg cumulative those). A total of 7,940 cardiovascular events happened during periods of nonexposure.

## Cumulative probabilities of cardiovascular diseases

The cumulative incidence estimates of all-cause CVD at 1 year increased from 1.4% (95% CI 1.4% to 1.5%) for periods of non-use, through 3.8% (95% CI 3.3% to 4.2%) for <5 mg, to 8.9% (95% CI 7.4% to 10.4%) for ≥25.0 mg daily dose; and were 1.6% (95% CI 1.4% to 1.7%) for unexposed periods and 1.4% (95% CI 1.0% to 1.9%) for ≥7,300 mg cumulative dose (Table 2 and Table E in S1 Table). We found higher dose–response estimates in men than in women and for atrial fibrillation and heart failure compared to other types of CVD (Table E, F, and G in S1 Table).

## Relationship between glucocorticoid dose and cardiovascular diseases

The increase in the hazard of all-cause CVD per 5 mg increase in daily dose was 1.08 (95% CI 1.07 to 1.10 per 5 mg/day), ranging from 1.07 (95% CI 1.06 to 1.09) for inflammatory bowel disease to 1.30 (95% CI 1.22 to 1.38) for systemic lupus erythematosus (Table 3 and Table H in S1 Table). We found strong dose–response estimates for current daily doses of <5.0 mg for all immune-mediated diseases (hazard ratio (HR) = 1.74, 95% CI 1.64 to 1.84; range 1.52 for poly-myalgia and/or giant cell arteritis to 2.82 for systemic lupus erythematosus), for all cardiovas-cular outcomes, and for daily and cumulative dose (Figs 1–3, Fig B–G in S1 Fig, and Fig A–F in S2 Fig). The highest glucocorticoid dose–response estimates were for heart failure and for acute myocardial infarction. We found similar patterns in prespecified sensitivity analyses, including restriction to patients with complete covariate data (Table I–M in S1 Table). Daily and cumulative dose–response estimates were generally higher among patients with longer underlying inflammatory disease duration and in those newly diagnosed (Table N–Q in S1 Table). Further, adjustment for the level of disease activity generally decreased the dose–response estimates, but associations remained statistically significant (Table R and S in S1 Table). In additional sensitivity analyses performed in response to peer reviewer comments, we found slightly higher dose–response estimates for glucocorticoid ever use when longer lengths of glucocorticoid exposure before the start of follow-up were considered and similar estimates for cumulative dose (Table T in S1 Table). We also found materially unchanged esti-mates after further adjusting for propensity scores for glucocorticoid prescribing (Table U in S1 Table).

## Discussion

In this longitudinal study of 87,794 adults diagnosed with at least 1 of 6 common immune-mediated inflammatory diseases, we quantified oral glucocorticoid dose-dependent risks of all-cause and type-specific CVDs taking into account changes in prescribed medication over time. At 1 year, the cumulative risk of all-cause CVD increased from 1.5% during periods with-out medication, through 3.8% for a daily prednisolone-equivalent dose <5 mg, to 9.1% for periods with a daily dose of ≥25.0 mg. We found strong dose-dependent increases in hazards of all-cause CVD, atherosclerotic diseases, heart failure, atrial fibrillation, and abdominal aor-tic aneurysm, regardless of the underlying immune-mediated disease, its activity, and dura-tion. The cardiovascular risk profile of the study patients showed high prevalence of modifiable risk factors, including current smoking (24.2% of patients), BMI ≥30 kg/m$^2$ (24.5%), and hypertension (25.1%).

**Table 2. Cumulative incidence estimates of CVDs per level of current daily and cumulative oral glucocorticoid PED.**

| Drug exposure variable | Cumulative probability, % (95% CI)[*] |
|---|---|
| Incident CVD, *n* (%) | 13,426 (15.3%) |
| **At 1 year** | 2.4 (2.3–2.5) |
| **Current daily PED, mg** | |
| non-use | 1.4 (1.4–1.5) |
| >0.0–4.9 | 3.8 (3.3–4.2) |
| 5.0–14.9 mg | 4.8 (4.4–5.1) |
| 15.0–24.9 mg | 7.2 (6.1–8.3) |
| ≥25.0 mg | 8.9 (7.4–10.4) |
| **Total cumulative PED in last year, mg** | |
| non-use | 1.6 (1.4–1.7) |
| >0.0–959.9 | 4.0 (3.6–4.3) |
| 960.0–3,054.9 | 4.4 (4.1–4.8) |
| 3,055.0–7,299.9 | 2.1 (1.9–2.3) |
| ≥7,300.0 | 1.4 (1.0–1.9) |
| **At 5 years** | 10.3 (10.0–10.5) |
| **Current daily PED, mg** | |
| non-use | 7.1 (6.9–7.3) |
| >0.0–4.9 | 19.7 (18.5–20.9) |
| 5.0–14.9 | 21.6 (20.7–22.5) |
| 15.0–24.9 | 26.8 (24.2–29.2) |
| ≥25.0 | 28.0 (25.1–30.7) |
| **Total cumulative PED, mg** | |
| non-use | 7.4 (7.1–7.7) |
| >0.0–959.9 | 9.7 (9.1–10.3) |
| 960.0–3,054.9 | 13.9 (13.3–14.6) |
| 3,055.0–7,299.9 | 15.4 (14.7–16.1) |
| ≥7,300.0 | 9.7 (9.1–10.3) |
| **At 10 years** | 19.1 (18.7–19.4) |
| **Current daily PED, mg** | |
| non-use | 14.6 (14.2–14.9) |
| >0.0–4.9 | 33.0 (31.4–34.6) |
| 5.0–14.9 | 35.1 (33.9–36.2) |
| 15.0–24.9 | 42.5 (38.8–45.9) |
| ≥25.0 | 39.9 (36.3–43.1) |
| **Total cumulative PED, mg** | |
| non-use | 14.5 (14.0–14.9) |
| >0.0–959.9 | 17.6 (16.7–18.5) |
| 960.0–3,054.9 | 20.8 (20.0–21.7) |
| 3,055.0–7,299.9 | 24.7 (23.8–25.6) |
| ≥7,300.0 | 21.9 (21.1–22.7) |

CI, confidence interval; CVDs, cardiovascular diseases; PED, prednisolone-equivalent dose.

[*]Unless stated otherwise. Number of patients at risk at 1, 5, and 10 years were 77,274, 44,460, and 19,562, respectively.

**Table 3. Associations between time-variant oral glucocorticoid PED and incident all-cause CVDs by immune-mediated inflammatory disease.**

| | Adjusted HRs with 95% CI | | | | | |
|---|---|---|---|---|---|---|
| | **All immune-mediated diseases**[*] | **Polymyalgia rheumatica and/or giant cell arteritis** | **Inflammatory bowel disease** | **Rheumatoid arthritis** | **Systemic lupus erythematosus** | **Vasculitis** |
| **No. of events** | 13,426 | 6,267 | 1,937 | 4,236 | 375 | 611 |
| **Ever use** (ref: non-use since 1 year prior to follow-up start) | 1.46 (1.40–1.53) | 1.25 (1.15–1.37) | 1.39 (1.26–1.52) | 1.63 (1.52–1.73) | 1.69 (1.35–2.12) | 1.52 (1.28–1.81) |
| **Current use** (ref: non-use) | 1.95 (1.87–2.02) | 1.69 (1.60–1.78) | 2.71 (2.43–3.02) | 2.11 (1.98–2.25) | 2.56 (2.02–3.25) | 2.07 (1.71–2.50) |
| **Current daily dose per 5 mg/day** | 1.09 (1.07–1.11) | 1.17 (1.15–1.19) | 1.08 (1.06–1.09) | 1.28 (1.25–1.31) | 1.28 (1.20–1.37) | 1.19 (1.13–1.25) |
| **Current daily dose** (ref: non-use) | 1.00 | 1.00 | 1.00 | 1.00 | 1.00 | 1.00 |
| 1–4.9 mg | 1.69 (1.57–1.81) | 1.50 (1.37–1.64) | 2.16 (1.66–2.82) | 1.84 (1.62–2.10) | 2.81 (1.92–4.11) | 1.92 (1.35–2.74) |
| 5.0–14.9 mg | 1.89 (1.81–1.98) | 1.70 (1.59–1.82) | 2.39 (2.03–2.82) | 2.00 (1.85–2.15) | 2.19 (1.62–2.94) | 1.92 (1.51–2.44) |
| 15.0–24.9 mg | 2.38 (2.13–2.67) | 2.07 (1.80–2.38) | 3.44 (2.49–4.74) | 2.79 (2.21–3.51) | 2.61 (1.15–5.94) | 2.46 (1.53–3.96) |
| ≥25 mg | 3.64 (3.28–4.04) | 2.76 (2.30–3.32) | 4.35 (3.43–5.52) | 4.98 (4.11–6.03) | 5.66 (3.20–10.01) | 3.07 (1.92–4.91) |
| **Total cumulative dose per 1,000 mg** | 1.01 (1.01–1.01) | 1.02 (1.01–1.02) | 1.01 (1.01–1.01) | 1.02 (1.02–1.03) | 1.03 (1.01–1.04) | 1.03 (1.02–1.04) |
| **Total cumulative dose** (ref: non-use) | 1.00 | 1.00 | 1.00 | 1.00 | 1.00 | 1.00 |
| 1–959.9 mg | 1.37 (1.29–1.45) | 1.24 (1.11–1.38) | 1.21 (1.06–1.38) | 1.47 (1.34–1.61) | 1.65 (1.20–2.26) | 1.54 (1.23–1.93) |
| 960–3,054.9 mg | 1.35 (1.28–1.43) | 1.14 (1.04–1.26) | 1.35 (1.18–1.54) | 1.52 (1.36–1.68) | 1.40 (0.95–2.06) | 1.31 (0.95–1.82) |
| 3,055–7,299.9 mg | 1.44 (1.36–1.52) | 1.21 (1.10–1.33) | 1.45 (1.24–1.69) | 1.72 (1.55–1.90) | 1.60 (1.10–2.34) | 1.27 (0.94–1.70) |
| ≥7,300 mg | 1.76 (1.66–1.86) | 1.50 (1.36–1.66) | 1.85 (1.58–2.16) | 1.80 (1.65–1.97) | 2.09 (1.53–2.85) | 1.91 (1.49–2.45) |

BMI, body mass index; CI, confidence interval; CVDs, cardiovascular diseases; HDL-cholesterol, high-density lipoprotein-cholesterol; HRs, hazard ratios; LDL-cholesterol, low-density lipoprotein-cholesterol; PED, prednisolone-equivalent dose.

Hazard ratios from Cox proportional imputed models adjusted for baseline age, sex, index of multiple deprivation, smoking status, ethnicity, BMI, comorbidities (diabetes, diagnosed hypertension, cancer, asthma, chronic obstructive pulmonary disease, and renal disease), biomarkers (total cholesterol, HDL-cholesterol, LDL-cholesterol, c-reactive protein, and creatinine), number of hospital admissions in last year, and prescribed non-oral glucocorticoids; and time-variant use of disease-modifying antirheumatic drugs and nonsteroidal anti-inflammatory drugs; the practice identifier was included as a random intercept to account for clustering effect.

[*]These estimates were additionally adjusted for the type of immune-mediated inflammatory disease diagnosed.

Previous studies have reported increased risk of composites of cardiovascular [2,3,11,12] or coronary heart disease [1], myocardial infarction [2,3,11,14,15], heart failure [2,3,11], stroke [2,3,16], and atrial fibrillation [7,13] in current glucocorticoid users. Some found an increased risk of CVD only for daily doses of 5 to 10 mg or higher [13,16,17]. Estimates from previous studies were based on current, baseline medication use [2,3,7,11–16] or dose, or average glucocorticoid dose in the last 6 to 12 months [1,3], without consideration of previously administered doses and changes in dose or medication use over time. Consistent with our findings, other studies assessing the relationship between glucocorticoid use and the risk of different types of CVDs reported stronger associations for heart failure than for other cardiovascular outcomes [2,3]. A summary of the methodology and major findings of previous studies is presented in our data supplement (Table V in S1 Table). This illustrates that our study is

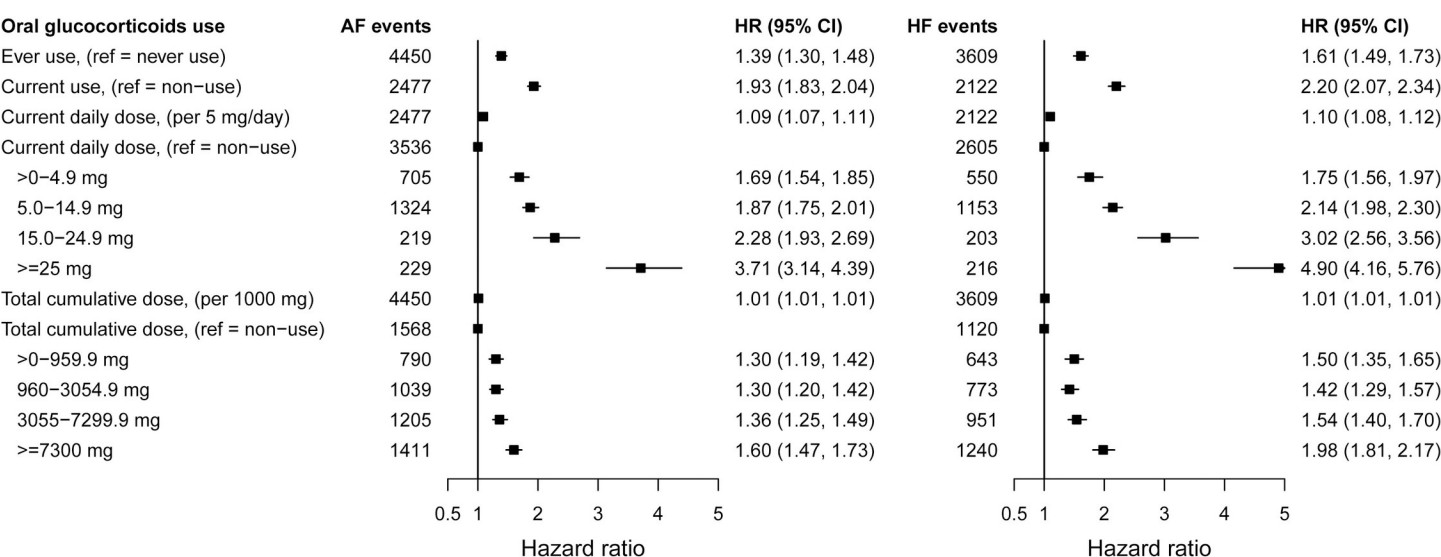

**Fig 1. Associations between time-variant oral glucocorticoid prednisolone-equivalent dose and incident atrial fibrillation and heart failure for patients with 6 immune-mediated inflammatory diseases.** HRs from Cox proportional imputed models adjusted for baseline age, sex, index of multiple deprivation, smoking status, ethnicity, BMI, type of immune-mediated inflammatory disease, comorbidities (diabetes, diagnosed hypertension, cancer, asthma, chronic obstructive pulmonary disease, and renal disease), biomarkers (total cholesterol, high-density lipoprotein cholesterol, low-density lipoprotein cholesterol, c-reactive protein, and creatinine), number of hospital admissions in last year, and prescribed non-oral glucocorticoids; and time-variant use of disease-modifying antirheumatic drugs and nonsteroidal anti-inflammatory drugs; the practice identifier was included as a random intercept to account for clustering effect. AF, atrial fibrillation; BMI, body mass index; CI, confidence interval; HF, heart failure; HR, hazard ratio.

substantially larger than most published work, offers estimates of risk for previously undefined categories of CVD events (e.g., abdominal aortic aneurysm and peripheral arterial disease), includes previously neglected immune-mediated inflammatory diseases (e.g., giant cell

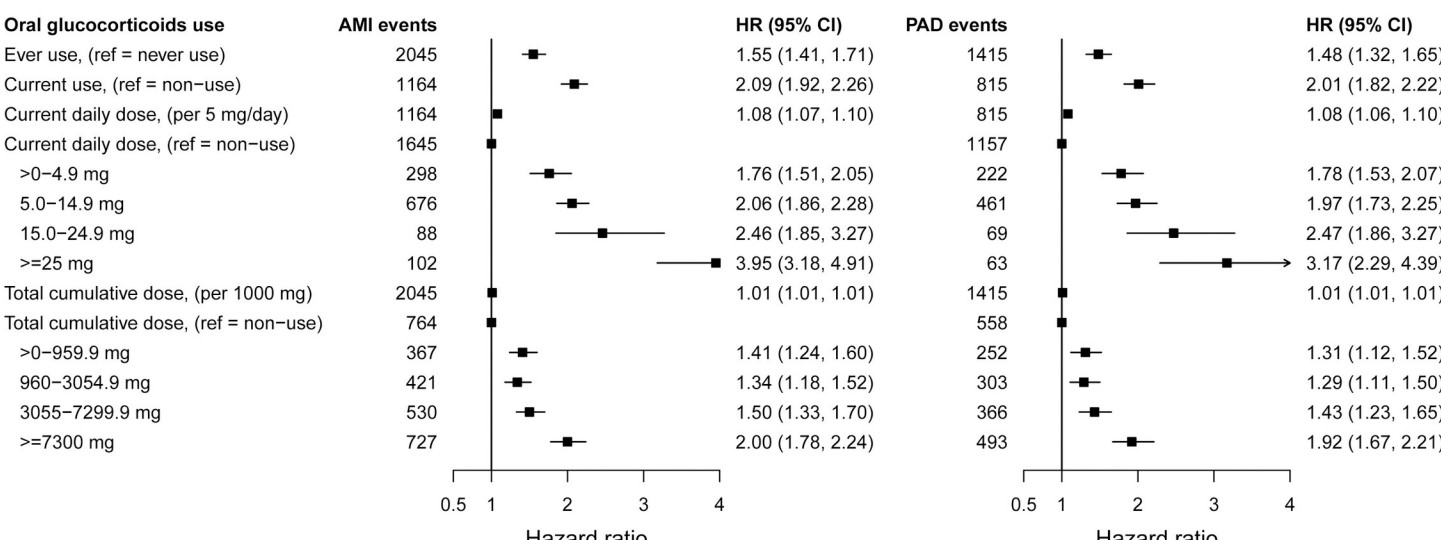

**Fig 2. Associations between time-variant oral glucocorticoid prednisolone-equivalent dose and incident AMI and PAD for patients with 6 immune-mediated inflammatory diseases.** HRs from Cox proportional imputed models adjusted for baseline age, sex, index of multiple deprivation, smoking status, ethnicity, BMI, type of immune-mediated inflammatory disease, comorbidities (diabetes, diagnosed hypertension, cancer, asthma, chronic obstructive pulmonary disease, and renal disease), biomarkers (total cholesterol, high-density lipoprotein cholesterol, low-density lipoprotein cholesterol, c-reactive protein, and creatinine), number of hospital admissions in last year, and prescribed non-oral glucocorticoids; and time-variant use of disease-modifying antirheumatic drugs and nonsteroidal anti-inflammatory drugs; the practice identifier was included as a random intercept to account for clustering effect. AMI, acute myocardial infarction; BMI, body mass index; CI, confidence interval; HR, hazard ratio; PAD, peripheral arterial disease.

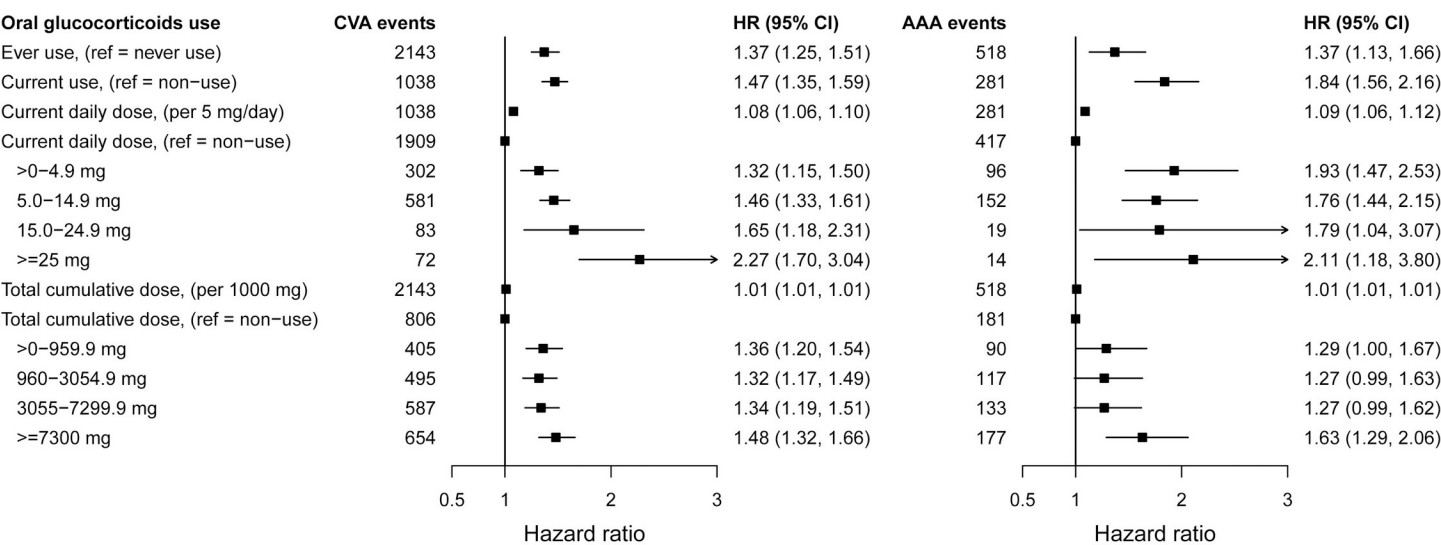

| Oral glucocorticoids use | CVA events | HR (95% CI) | AAA events | HR (95% CI) |
|---|---|---|---|---|
| Ever use, (ref = never use) | 2143 | 1.37 (1.25, 1.51) | 518 | 1.37 (1.13, 1.66) |
| Current use, (ref = non−use) | 1038 | 1.47 (1.35, 1.59) | 281 | 1.84 (1.56, 2.16) |
| Current daily dose, (per 5 mg/day) | 1038 | 1.08 (1.06, 1.10) | 281 | 1.09 (1.06, 1.12) |
| Current daily dose, (ref = non−use) | 1909 | | 417 | |
| >0−4.9 mg | 302 | 1.32 (1.15, 1.50) | 96 | 1.93 (1.47, 2.53) |
| 5.0−14.9 mg | 581 | 1.46 (1.33, 1.61) | 152 | 1.76 (1.44, 2.15) |
| 15.0−24.9 mg | 83 | 1.65 (1.18, 2.31) | 19 | 1.79 (1.04, 3.07) |
| >=25 mg | 72 | 2.27 (1.70, 3.04) | 14 | 2.11 (1.18, 3.80) |
| Total cumulative dose, (per 1000 mg) | 2143 | 1.01 (1.01, 1.01) | 518 | 1.01 (1.01, 1.01) |
| Total cumulative dose, (ref = non−use) | 806 | | 181 | |
| >0−959.9 mg | 405 | 1.36 (1.20, 1.54) | 90 | 1.29 (1.00, 1.67) |
| 960−3054.9 mg | 495 | 1.32 (1.17, 1.49) | 117 | 1.27 (0.99, 1.63) |
| 3055−7299.9 mg | 587 | 1.34 (1.19, 1.51) | 133 | 1.27 (0.99, 1.62) |
| >=7300 mg | 654 | 1.48 (1.32, 1.66) | 177 | 1.63 (1.29, 2.06) |

**Fig 3. Associations between time-variant oral glucocorticoid prednisolone-equivalent dose and incident cerebrovascular disease and AAA for patients with 6 immune-mediated inflammatory diseases.** HRs from Cox proportional imputed models adjusted for baseline age, sex, index of multiple deprivation, smoking status, ethnicity, BMI, type of immune-mediated inflammatory disease, comorbidities (diabetes, diagnosed hypertension, cancer, asthma, chronic obstructive pulmonary disease, and renal disease), biomarkers (total cholesterol, high-density lipoprotein cholesterol, low-density lipoprotein cholesterol, c-reactive protein, and creatinine), number of hospital admissions in last year, and prescribed non-oral glucocorticoids; and time-variant use of disease-modifying antirheumatic drugs and nonsteroidal anti-inflammatory drugs; the practice identifier was included as a random intercept to account for clustering effect. AAA, abdominal aortic aneurysm; BMI, body mass index; CI, confidence interval; CVA, cerebrovascular disease; HR, hazard ratio.

arteritis), and provides more detailed quantification of glucocorticoid use. Where analysis groups are similar, the 95% CI of our estimates overlap with those reported in previous studies in most cases, although our intervals are smaller due to the larger size of our cohort.

The elevated absolute risk of CVD in patients receiving high doses of glucocorticoids, of similar magnitude to that of patients with diabetes or established CVD, warrants the need to implement and evaluate intensive lifestyle modification interventions to this high-risk group. The dose-dependent increased risk of CVDs, including atherosclerotic diseases, heart failure, and atrial fibrillation observed in our study, supports the need for close monitoring of cardiovascular risk in patients diagnosed with immune-mediated inflammatory diseases during glucocorticoid treatment and in the period after therapy discontinuation. Of cardiovascular risk scores currently used to guide decision on when to start primary cardiovascular prophylaxis, only the QRISK3 [32] considers whether the patient is currently taking glucocorticoids (as a binary "yes/no" predictor that ignores dose and recent exposure) and whether he/she is diagnosed with rheumatoid arthritis or systemic lupus erythematosus. Further refinements of this risk prediction tool, taking into account cumulative and/or current dose, might therefore improve its performance to identify patients in need for primary cardiovascular prevention. Our findings also emphasise the importance of rapid glucocorticoid dose tapering and discontinuation as soon as disease control is achieved, as well as the importance of evaluating the safety profile of alternative therapeutic options for patients with autoimmune-mediated inflammatory diseases.

This study has some key strengths. The estimation of drug dose–response risks in this population-based cohort of all people with immune-mediated inflammatory diseases with different levels of activity and duration minimised the introduction of selection bias and increased the generalisability of the results. The use of linked health data from primary care and hospital facilities and the mortality registry, and diagnostic codes extensively used and validated for

cardiovascular research [25–28,33,34], increased ascertainment of the study population and all the outcomes assessed. Estimates of positive predictive values reported in validation studies for the diseases of interest are ≥75%. Information on prescribed medication is prospectively collected and includes all prescriptions issued in primary care, where patients with immune-mediated diseases are primarily treated. We derived the dose of oral glucocorticoids and the duration of prescribed medication from the directions given to patients on how to take their treatment. During periods of dose tapering, when these directions were unspecific (e.g., written "as directed"), we used the longitudinal doses prescribed to the patients to impute the dose taken. We minimised time-related bias through use of time-variant medication variables (both exposure and confounders) and a start of patient follow-up that was unrelated to the start or use of glucocorticoid therapy. We adjusted HRs for established cardiovascular risk factors (e.g., smoking, hypertension, and diabetes), concomitant use of medications (e.g., time-variant disease-modifying antirheumatic drugs, nonsteroidal anti-inflammatory drugs, and non-oral glucocorticoid use). In the primary analysis, we used multiple imputation to handle missing baseline biomarkers and smoking data. We found similar patterns of dose–response in sensitivity analyses, including analyses in which we used a separate category for missing covariate data, in those restricted to individuals with complete covariate data, and in those unadjusted for baseline biomarker information with high level of missingness (>60% of patients).

However, this study also has limitations. The lack of data on hospital prescribed medication and on drug adherence is likely to have resulted in underestimation of the dose taken when specialists treated the patients and might have overestimated the dose taken in periods of low disease activity for some patients. The resulting misclassification is likely to have reduced the size of dose–response estimates. Furthermore, although the main purpose of the study was to provide estimates of oral glucocorticoid dose–response for patients with the inflammatory diseases studied without making aetiological inferences, associations might be confounded by indication of glucocorticoid therapy and affected by unmeasured confounding. We therefore examined the effect of confounding by indication and ascertainment bias through adjustment by periods of disease activity and disease-modifying antirheumatic drugs use and performing analyses according to duration of the underlying disease. The resulting dose–response associations remained strong and statistically significant, and estimates obtained when propensity score model adjustment was used to balance drug exposure groups did not change.

In conclusion, we reported improved estimates of dose-dependent risks of CVDs. Our findings highlight the importance of implementing and evaluating targeted intensive cardiovascular risk factor modification interventions; promptly and regularly monitor patient cardiovascular risk, beyond diagnosis of inflammatory arthropathies and systemic lupus erythematosus, even when prescribing low prednisolone-equivalent doses; and the need for refining existing risk prediction tools for primary prevention of CVDs. Furthermore, the estimates of risk can be used to conduct cost-effectiveness and benefit–harm evaluations that guide the introduction of newly licenced glucocorticoid-sparing drugs for the treatment of immune-mediated inflammatory diseases. These estimates are to be complemented by future work on the estimation of risk of cardiovascular events beyond the first occurrence of CVD that are considered in calculations of glucocorticoid-associated costs.

## Disclaimer

The views expressed are those of the author(s) and not necessarily those of the NHS, the NIHR, or the Department of Health and Social Care. The study funders had no role in the study design, data collection, analysis or interpretation, in the writing of the paper, or in the decision to submit the paper for publication.

## Supporting information

**S1 Text. Supplementary methods including: Sources of data; Covariate definition; Definition of flare; Multiple imputation of glucocorticoid dose and covariates; and Propensity score for prescribing information.**
(DOCX)

**S1 Table.** Table A. Definition of immune-mediated inflammatory diseases by data source. Table B. Prednisolone-equivalent dose conversion factors for glucocorticoids. Table C. Definition of cardiovascular outcomes by data source. Table D. Observation time and incidence rates of cardiovascular diseases by sex. Table E. Cumulative incidence estimates of cardiovascular diseases per level of current daily and cumulative oral glucocorticoid prednisolone-equivalent dose by type of immune-mediated inflammatory disease. Table F. Cumulative incidence estimates of cardiovascular diseases per level of current daily and cumulative oral glucocorticoid prednisolone-equivalent dose by type of immune-mediated inflammatory disease in men. Table G. Cumulative incidence estimates of cardiovascular diseases per level of current daily and cumulative oral glucocorticoid prednisolone-equivalent dose in women. Table H. Associations between time-variant oral glucocorticoid prednisolone-equivalent dose and incident all-cause cardiovascular disease by immune-mediated inflammatory disease, reported as crude hazard ratios with 95% CI. Table I. Association between time-variant oral glucocorticoid dose and incident cardiovascular disease in patients with 6 immune-mediated inflammatory diseases from complete case analysis. Table J. Association between time-variant oral glucocorticoid dose and incident all-cause cardiovascular disease by type of immune-mediated inflammatory disease from analysis in which missing covariate values were coded as a separate category. Table K. Association between time-variant oral glucocorticoid dose and incident all-cause cardiovascular disease by type of immune-mediated inflammatory disease from analysis in which biomarkers with over 60% missing data were excluded. Table L. Association between time-variant oral glucocorticoid dose and 6 incident cardiovascular diseases in patients with 6 immune-mediated inflammatory diseases from analysis in which missing covariate values were coded as a separate category. Table M. Association between time-variant oral glucocorticoid dose and 6 incident cardiovascular disease in patients with 6 immune-mediated inflammatory diseases from analysis in which biomarkers with over 60% missing data were excluded. Table N. Association between time-variant oral glucocorticoid dose and incident all-cause cardiovascular disease by type of immune-mediated inflammatory disease, restricted to patients with newly diagnosed immune-mediated inflammatory disease. Table O. Association between time-variant oral glucocorticoid dose and 6 incident cardiovascular diseases in patients with 6 immune-mediated inflammatory diseases, restricted to patients with newly diagnosed immune-mediated inflammatory disease. Table P. Association between time-variant oral glucocorticoid dose and 6 incident cardiovascular diseases in patients with 6 immune-mediated inflammatory diseases, restricted to patients diagnosed with immune-mediated inflammatory disease within 2 years. Table Q. Association between time-variant oral glucocorticoid dose and 6 incident cardiovascular diseases in patients with 6 immune-mediated inflammatory diseases, restricted to patients diagnosed with immune-mediated inflammatory diseases for over 2 years. Table R. Association between time-variant oral glucocorticoid dose and 6 incident cardiovascular diseases in patients with 6 immune-mediated inflammatory diseases, adjusted for periods of flare during follow-up (defined by biomarker or 5-mg daily dose increase). Table S. Association between time-variant oral glucocorticoid dose and 6 incident cardiovascular diseases in patients with 6 immune-mediated inflammatory diseases, adjusted for periods of flare during follow-up (defined by biomarker or 10-mg daily dose increase). Table T. Associations between time-variant oral glucocorticoid prednisolone-equivalent dose and incident all-cause

cardiovascular disease by immune-mediated inflammatory disease, according to the number of years of exposure considered prior to follow-up start [Additional sensitivity analysis]. Table U. Associations between time-variant oral glucocorticoid prednisolone-equivalent dose and incident all-cause cardiovascular disease by immune-mediated inflammatory disease, adjusted for propensity score for prescribing indication [Additional sensitivity analysis]. Table V. Summary of the methodology and major findings of previous studies investigating the association between glucocorticoid dose and cardiovascular diseases.
(DOCX)

**S1 Fig.** Fig A. Flow diagram of the study cohort. Fig B. Association between time-variant oral glucocorticoid dose and incident atrial fibrillation and heart failure in patients with polymyalgia rheumatica and/or giant cell arteritis. Fig C. Association between time-variant oral glucocorticoid dose and incident acute myocardial infarction and peripheral arterial disease in patients with polymyalgia rheumatica and/or giant cell arteritis. Fig D. Association between time-variant oral glucocorticoid dose and incident cerebrovascular disease and abdominal aortic aneurysm in patients with polymyalgia rheumatica and/or giant cell arteritis. Fig E. Association between time-variant oral glucocorticoid dose and incident atrial fibrillation and heart failure in patients with rheumatoid arthritis. Fig F. Association between time-variant oral glucocorticoid dose and incident acute myocardial infarction and peripheral arterial disease in patients with rheumatoid arthritis. Fig G. Association between time-variant oral glucocorticoid dose and incident cerebrovascular disease and abdominal aortic aneurysm in patients with rheumatoid arthritis
(DOCX)

**S2 Fig.** Fig A. Association between time-variant oral glucocorticoid dose and incident atrial fibrillation and heart failure in patients with inflammatory bowel disease. Fig B. Association between time-variant oral glucocorticoid dose and incident acute myocardial infarction and peripheral arterial disease in patients with inflammatory bowel disease. Fig C. Association between time-variant oral glucocorticoid dose and incident cerebrovascular disease and abdominal aortic aneurysm in patients with inflammatory bowel disease. Fig D. Association between time-variant oral glucocorticoid dose and incident atrial fibrillation and heart failure in patients with vasculitis. Fig E. Association between time-variant oral glucocorticoid dose and incident acute myocardial infarction and peripheral arterial disease in patients with vasculitis. Fig F. Association between time-variant oral glucocorticoid dose and incident cerebrovascular disease and abdominal aortic aneurysm in patients with vasculitis.
(DOCX)

**S1 RECORD-PE Checklist.**
(DOCX)

## Author Contributions

**Conceptualization:** Mar Pujades-Rodriguez.

**Data curation:** Mar Pujades-Rodriguez.

**Formal analysis:** Jianhua Wu.

**Funding acquisition:** Mar Pujades-Rodriguez.

**Methodology:** Mar Pujades-Rodriguez, Jianhua Wu.

**Project administration:** Mar Pujades-Rodriguez.

**Supervision:** Mar Pujades-Rodriguez.

**Writing – original draft:** Mar Pujades-Rodriguez.

**Writing – review & editing:** Ann W. Morgan, Richard M. Cubbon, Jianhua Wu.

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
