## [Decision Letter · Decision Letter 0]

2 Jul 2020

Dear Dr. Pujades-Rodriguez,

Thank you very much for submitting your manuscript "Dose-dependent oral glucocorticoid cardiovascular risk in people with immune-mediated inflammatory diseases" (PMEDICINE-D-19-02196) for consideration at PLOS Medicine; sincere apologies for the delay in taking the paper through peer review and to an initial decision point. 

[LINK]

In light of these reviews, I am afraid that we will not be able to accept the manuscript for publication in the journal in its current form, but we would like to consider a revised version that addresses the reviewers' and editors' comments. Obviously we cannot make any decision about publication until we have seen the revised manuscript and your response, and we plan to seek re-review by one or more of the reviewers. 

We expect to receive your revised manuscript by Jul 23 2020 11:59PM. Please email us (plosmedicine@plos.org) if you have any questions or concerns.

We look forward to receiving your revised manuscript. 

Sincerely,

Emma Veitch, PhD

PLOS Medicine

On behalf of Thomas McBride, PhD

Senior Editor 

PLOS Medicine

plosmedicine.org

*We'd suggest revising the title into PLOS Medicine's usual style, this should have the study design (eg "A randomized controlled trial," "A retrospective study," "A modelling study," etc.) in the subtitle after a colon. 

*Please redraft parts of the abstract so that this uses complete sentence structure rather than fragments (ie, "**WE AIMED** to quantify glucocorticoid dose-dependent cardiovascular risk.." etc).

*In the last sentence of the Abstract Methods and Findings section, please include a brief summary of any key limitations of the study's methodology.

*At this stage, we ask that you include a short, non-technical Author Summary of your research to make findings accessible to a wide audience that includes both scientists and non-scientists. The Author Summary should immediately follow the Abstract in your revised manuscript. This text is subject to editorial change and should be distinct from the scientific abstract. Please see our author guidelines for more information: https://journals.plos.org/plosmedicine/s/revising-your-manuscript#loc-author-summary

*The authors might like to consider using an appropriate reporting guideline to enhance the detail of reporting of key aspects of the study methods and findings. Options might include RECORD (https://www.equator-network.org/reporting-guidelines/record/ - designed for use with observational studies based on routinely collected data) or RECORD-PE (https://www.equator-network.org/reporting-guidelines/record-pe/ - RECORD for pharmacoepidemiology). If doing so the completed reporting guideline checklist should be appended as a supporting information file with the revised paper.

*Please clarify whether the analysis reported here corresponds to one laid out in a prospective protocol or analysis plan? Please state this (either way) early in the Methods section.

Comments from the academic editor:

I would also like the authors to discuss the quantitative difference between some of the earlier works and this paper (perhaps in a supplementary table). In addition, given that one of the key rationales for this paper is incomplete confounder adjustment in previous papers, it might be good to provide a table reporting effects by cumulative adjustment of various confounders. Does it make any difference? 

Comments from the reviewers:

Reviewer #1: The authors of this paper report the results of an observational study using a UK CPRD electronic medical records linked to hospital episodes statistics, which evaluated dosage-dependent relationship of oral glucocorticoids prescribing and risk of all-cause CVD outcomes as cause-specific CVD six immune-related inflammatory diseases. They report dose-dependent estimates which increased risk of CVD significantly compared to non-use for all the inflammatory diseases, with highest estimates for heart failure and myocardial infarction. 

The paper results largely confirm what has been already been extensively reported in various observational studies on the association between oral glucocorticoids and CVD. The study does goes further than previous studies to examine cumulative prescribing over time, with these results are largely consistent with other studies. 

I have reviewed the statistical methods and the models used are appropriate, using multivariate Cox models and KM techniques. The authors had also conducted a complete case analysis and compared this to multiple imputation which is good practice. However, there are some suggestions/comments I have on the study design. 

Major: 

(1) Relating to exposure period of only one year prior to index date and residual confounding

In pharmaco-epidemiological studies, the major issue here is consideration for protopathic bias and confounding by indication. In this study, neither issue is sufficiently investigated for or considered in the study design. See https://www.ncbi.nlm.nih.gov/pmc/articles/PMC4594717/

Firstly, to control for protopathic bias (as the outcome CVD here is associated both with the exposure and the indication for the exposure, symptoms or diagnosis itself, usually lag time can be added to exposure period (exclusion of any drug prescribing in the year before the index date). Often times, additional sensitivity analyses are also incorporated at longer exclusions periods for prescribing prior to the index date (e.g. > 2 years or > 5 years). This allows testing of the robustness of the primary results. If indeed, there is a dose-response relationship, then this effect should remain in the lag-time analyses. The present analyses uses only an exposure period of one year prior to diagnosis.

Related to this, confounding by indication will therefore also be a potential issue here as well. The indication for prescriptions itself (diagnosis of inflammatory diseases) is related to the outcome of CVD here. This is going to be the case here as the inflammatory diseases are known to be associated with an increased risk of CVD, with more severe symptoms also potentially being treated more aggressively. This creates residual confounding which is not appropriately taken into account. 

To account for methods which can minimize this: (1) using a propensity scores to adjust the statistical models - a statistical way to try to achieve balance between exposure groups in observational research designs or (2) using a comparator group where the indication for the prescription is not related to the outcome (i.e. a disease group which would be prescribed oral glucocorticoids). Covariate adjustment as completed in this analysis adjusts for each individual variable as a potential confounder but not achieve as robust adjustments for residual confounding as in propensity score adjustments (the results can differ based on these methods). See https://www.bmj.com/content/347/bmj.f6409.full

These additional analyses would test the robustness of the primary results and should be considered. In general (especially using observational or real-world evidence data), the burden is up to the researchers to provide additional analyses to test the robustness of the results under various assumptions. The study here in my opinion is incomplete as issues related to residual confounding have not been considered. 

(2) Eligibility of patients: It's not clear patients or mentioned if patients have more than one inflammatory condition which group they would fall under - I would have to presume that the groups are not mutually exclusive (i.e. an individual with co-morbid inflammatory conditions) could be contributing to more than one category. In the sub-analyses of each disease, presenting the results separately is not an issue, but in the combined analysis, presumably this could potentially double count individuals who may fall into more than one category? Is this is the case, then overall effect estimates would could potentially be exaggerated. Please clarify?

(3) Outcomes: Cause-specific incidence outcomes are fine but combining composite CVD outcomes presents an issue of competing risk events (as the individuals may have multiple types of events). Using a cumulative incidence model would be more helpful when presenting the overall composite outcome. 

Minor: 

Lines 109: It's not clear here what the authors mean by follow-up started when they "first became eligible". Is this the date which first instance of a diagnosis of any of the six immune-mediated inflammatory disease? Please clarify

Line 120: Conversion to prednisolone-equivalent dose. Another common way to standardise the different drug dosages is to use the WHO DDD as the benchmark. See https://www.whocc.no/atc_ddd_index/?code=H02AB. Many pharmaco-epidemiology studies use this approach for comparability. Perhaps authors could consider this as a sensitivity analyses. 

Limitation: limitation of CPRD data should be mentioned - covers only prescribing in general practice and does not cover any in-hospital prescribing associated with any acute events during the exposure or follow-up periods. 

Reviewer #2: This is a large population-based cohort study that examines glucocorticoid dose-dependent cardiovascular risk in six immune-mediated diseases. The study adds important knowledge in terms of dose-dependent associations and absolute risks. Further, the study has more sufficent confounder control than many prior observational studies and has made substantiel efforts to disentangle confoudning by indication and disease severity. The study is very comprehensive, however, I have comments that need to be adressed. 

I declare no conflicts of interest.

ABSTRACT

Comment #1

Page 2, lines 28-30: 

"Evidence for the association between glucocorticoid dose and cardiovascular risk is weak for moderate and low doses. To quantify glucocorticoid dose-dependent cardiovascular risk in people with six immune-mediated inflammatory diseases."

It is difficult to understand if it is the evidence or the associations that are weak. Further, you may add "This study aimed" to the second sentence. 

Comment #2 

Page 2, lines 43-46.

"We found strong dose-dependent estimates for all immune-mediated diseases (hazard ratio [HR] for <5.0mg daily dose vs. non-use=1.74, 95%CI: 1.64-1.84; range 1.52 for polymyalgia rheumatic and/or giant cell arteritis to 2.82 for systemic lupus erythematosus), all cardiovascular outcomes, regardless of disease activity level."

This sentence is difficult to follow, you may consider to re-write. 

INTRODUCTION

I have three minor suggestions to the Introduction.

Comment #3

Page 4, lines 58-60:

"Patients with immune-mediated inflammatory diseases often receive long-term courses of oral glucocorticoids to reduce disease activity and inflammation during the initial episode and subsequent episodic flares." 

I suggest to delete the word "long-term" as it is not correct for all six diseases and also it depends on the definition of long-term treatment. 

It is accurate that long-term oral glucocorticoid treatment is indicated for polymyalgia rheumatica and giant cell arthritis. However, for rheumatoid arthritis long-term oral glucocorticoid treatment is rarely used and only if DMARDs cannot be used. Oral short-term/ medium-term glucocorticoid treatment may be used until effect of DMARDs. 

Comment #4

Page 4, lines 61- 65. The first sentence is in present tense ("can") and the last is in past tense ("could"). 

Comment #5

Page 4, lines 79-81:

"Our study aimed to estimate daily and cumulative dose-dependent oral glucocorticoid cardiovascular disease risk accurately in people diagnosed with six common immune-mediated inflammatory diseases in England."

I understand that the word "accurately" refers to the time-variant measure of the exposure, but I would delete/replace the word as it is may also refer to the quality of your exposure data as validity and completeness. 

METHODS

Comment #6

How was the distribution of the different generic types of oral glucocorticoids, i.e. the frequency of prednisolone use, betamethasone use etc.? 

Comment #7

From Table 2, S1 File it reads as if budesonide was included as an oral glucocorticoid? The main actions are in the mucosa (i.e. locally acting) and the bioavailability is < 20%. I would suggest not to include budesonide as an oral glucocorticoid exposure in your analyses or examine in a sensitivity analysis if not including budesonide as an expsoure changes your estimates. 

Comment #8

In Table 1 age and biomarkers are described by mean (SD). Are they normal distributed? Else, please change this to e.g. median. 

Comment #9

Were the biomarkers included as continous variables in the adjusted models and if so, how were they modeled (i.e. linear, cubic spline etc.) 

Comment #10

Page 7, lines 161-162.

"We estimated cumulative probabilities of CVD outcomes using Kaplan-Meier methods."

The cumulative incidences/absolute risks of CVD were estimated using the Kaplan-Meier estimator. As you have competing risk by death, please take this into account by computing the cumulative incidence function (e.g. the Aalen Johansen estimator) instead of the Kaplan-Meier function. The Kaplan-Meier function likely overestimates the cumulative incidence in competing risk settings (reference: Lacny et al. Kaplan-Meier survival analysis overestimates cumulative incidence of health-related events in competing risk settings: a meta-analysis. Journal of Clinical Epidemiology 93 (2018) 2535).

RESULTS

Comment #11

In Table 1 you have provided count and percentage of DMARDs ever during follow-up, which is relevant since you model this as a time-varying variable. Would it be relevant to do the same for NSAIDs? 

Comment #12

How many were loss to follow up (left the family practice)? 

Comment #13

Your absolute risk estimates are important from a clinical point of view, hence, I think some of the results from e.g. Table 5, S1 File deserve to be in the main text as a Table or Figure (if space enough and after re-estimating them in accordance with comment #7). 

Comment #14

It would be appropiate to show crude hazard ratios also (maybe just in the supplementary) 

DISCUSSION

Comment #15

Page 21:

"We adjusted estimates of risk for established cardiovascular risk factors (e.g. smoking, hypertension and diabetes), concomitant use of medications (e.g. time-variant disease-modifying antirheumatic drugs, non-steroidal antiinflammatory drugs, non-oral glucocorticoid use)"

The estimates of risk (absolute risk) were unadjusted, so please delete "estimates of risk" or replace with another term. 

Comment #16

It would strengthen the paper if estimates of e.g. PPV and completeness for the immune-mediated inflammatory diseases and cardiovascular outcomes were included in the discussion.

[LINK]

---

## [Decision Letter · Decision Letter 1]

2 Oct 2020

Dear Dr. Pujades-Rodriguez,

Thank you very much for re-submitting your manuscript "Dose-dependent oral glucocorticoid cardiovascular risk in people with immune-mediated inflammatory diseases" (PMEDICINE-D-19-02196R1) for review by PLOS Medicine.

I have discussed the paper with my colleagues and the academic editor and it was also seen again by two reviewers. I am pleased to say that provided the remaining editorial and production issues are dealt with we are planning to accept the paper for publication in the journal.

[LINK]

We look forward to receiving the revised manuscript by Oct 09 2020 11:59PM. 

Sincerely,

Thomas McBride, PhD

Senior Editor 

PLOS Medicine

plosmedicine.org

Requests from Editors:

1- Thank you for including your completed RECORD-PE checklist. Please update the file to replace the page numbers with paragraph numbers per section (e.g. "Methods, paragraph 1"), since the page numbers of the final published paper may be different from the page numbers in the current manuscript. Please also add the following statement, or similar, to the Methods: "This study is reported as per the reporting of studies conducted using observational routinely collected health data statement for pharmacoepidemiology (RECORD-PE) guideline (S1 Checklist)."

2- Thank you for noting in the Methods which analyses were pre-planned and which were in response to reviewers. Is it possible to provide a copy of the generic protocol written for ISAC as a supplementary file, or link to a published copy from the Methods section? Please also state in the Methods section that no specific protocol was written for the analyses of CVD risk.

3- Competing Interests: could you specify the research contract organization where MPR is employed?

4- Perhaps more accurate for the title to read *risks*.

5- Thank you for providing the website for access to the raw dataset. Please also provide any other information an interested researcher would need (e.g., accession number) to access this specific dataset.

6- The Abstract Background section could start with a sentence introducing the main indications for glucocorticoids ( immune-mediated inflammatory diseases) and how commonly they are used (similar to the first point of the Author Summary).

7- The Abstract Methods and Findings could note the proportion of person-years spent at each exposure level for glucocorticoids (or at least amount of time exposed and not exposed) during the years studied. 

8- The Abstract Methods and findings could list the HRs for type-specific CVDs rather than describing that ratios for HF and AMI were “higher”.

9- The first sentence of the Abstract Conclusions should state what you found (increased risk of CVD even at lower doses), without overreaching what can be concluded from the data; the phrase "In this study, we observed ..." may be useful.

10- Abstract Conclusions, second sentence: “*These* results highlight the importance of…”

11- Author Summary, line 83, if this refers to the HR of 1.74, please edit to read “... the overall risk of CVD *was elevated* for individuals...”

12- Line 88, please remove “strong”.

13- Discussion, page 28, please change the first sentence of the first full paragraph to “The *elevated absolute risk of CVD in patients receiving...”

14- Please include a paragraph in the Discussion (just before the Conclusions) on the limitations of this study. It would be useful to mention the possible issue of confounding by indication, and other unmeasured confounders too.

15- Please remove the spaces from the reference call-outs (e.g. “[2,3,11,12] ”).

16- Please edit the SI list at the end of the main text to include each of the supporting information items (text, tables, figures, and checklist). Callouts from the text should reference the specific SI item.

17- Reference 20: papers cannot be listed in the reference list until they have been accepted for publication or are publicly available on a preprint archive. If the reference is now published please update. If not, the information may be cited in the text as a personal communication with the author if the author provides written permission to be named. Alternatively please provide a different appropriate reference. 

Comments from Reviewers:

Reviewer #2: I recommend the article for publication. The authors have addressed all my comments very well.

Reviewer #3: I confine my remarks to statistical aspects of this paper.

The paper was earlier seen by a different statistician and those comments were addressed. I have no additional issues and I now recommend publication.

Peter Flom

[LINK]

---

## [Editor Report · Decision Letter 2]

29 Oct 2020

Dear Dr Pujades-Rodriguez, 

On behalf of my colleagues and the academic editor, Dr. Kazem Rahimi, I am delighted to inform you that your manuscript entitled "Dose-dependent oral glucocorticoid cardiovascular risks in people with immune-mediated inflammatory diseases: a population-based cohort study" (PMEDICINE-D-19-02196R2) has been accepted for publication in PLOS Medicine. 

PRODUCTION PROCESS

Before publication you will see the copyedited word document (within 5 business days) and a PDF proof shortly after that. The copyeditor will be in touch shortly before sending you the copyedited Word document. We will make some revisions at copyediting stage to conform to our general style, and for clarification. When you receive this version you should check and revise it very carefully, including figures, tables, references, and supporting information, because corrections at the next stage (proofs) will be strictly limited to (1) errors in author names or affiliations, (2) errors of scientific fact that would cause misunderstandings to readers, and (3) printer's (introduced) errors. Please return the copyedited file within 2 business days in order to ensure timely delivery of the PDF proof. 

If you are likely to be away when either this document or the proof is sent, please ensure we have contact information of a second person, as we will need you to respond quickly at each point. Given the disruptions resulting from the ongoing COVID-19 pandemic, there may be delays in the production process. We apologise in advance for any inconvenience caused and will do our best to minimize impact as far as possible.

PRESS

PROFILE INFORMATION

Thank you again for submitting the manuscript to PLOS Medicine. We look forward to publishing it. 

Best wishes, 

Thomas McBride, PhD

Senior Editor 

PLOS Medicine

plosmedicine.org